# Influence of Temperature, Photoperiod, and Supplementary Nutrition on the Development and Reproduction of *Scutellista caerulea* Fonscolombe (Hymenoptera: Pteromalidae)

**DOI:** 10.3390/insects14010082

**Published:** 2023-01-13

**Authors:** Xian Li, Zhengpei Ye, Junyu Chen, Junhong Zhu, Dongyin Han, Jianyun Wang, Lei Li, Yueguan Fu, Fangping Zhang

**Affiliations:** 1Environment and Plant Protection Institute, Chinese Academy of Tropical Agricultural Sciences, Haikou 570100, China; 2College of Plant Protection, Hainan University, Haikou 570100, China; 3Engineering Research Center for Biological Control of Tropical Crops Diseases and Insect Pests, Haikou 570100, China; 4Key Laboratory of Integrated Pest Management on Tropical Crops, Ministry of Agriculture and Rural Affairs, Haikou 570100, China; 5Hainan Key Laboratory for Monitoring and Control of Tropical Agricultural Pests, Haikou 570100, China

**Keywords:** parasitism rate, developmental duration, emergence, female ratio, life span

## Abstract

**Simple Summary:**

Parasitoids are the natural enemies of many pests. Using parasitoids is a valuable method for controlling pests. However, to effectively use parasitoids, it is necessary to understand their optimal living conditions. *Scutellista caerulea* Fonscolombe (Hymenoptera: Pteromalidae) is an important enemy of pestiferous scale, *Parasaissetia nigra* Nietner (Hemiptera: Coccidae). To identify the optimal conditions for the population growth of *S. caerulea*, we assessed how temperature, photoperiod, and supplementary nutrition affected its development and reproduction. Our results revealed that the most suitable conditions for the population growth of *S. caerulea* was at 30 to 33 °C, with 12 to 14 h of daily light, and the provision of sucrose or honey as supplemental diet. These results provide a reference for the indoor rearing of *S. caerulea*.

**Abstract:**

*Scutellista ciruela* Fonscolombe has a significant controlling effect on the rubber tree pest, *Parasaissetia nigra* Nietner. To identify the optimal conditions for the population growth of *S. caerulea*, we assessed how temperature, photoperiod, and supplementary nutrition affected its development and reproduction. The results demonstrated that the number of eggs laid and parasitism rates of *S. caerulea* were the highest at 33 °C. The developmental rate of *S. caerulea* was the fastest and the number of emerged adults the highest. The number of eggs laid and the parasitism rates increased when the light duration increased within a day. Females did not lay any eggs when the whole day was dark. At a photoperiod of 14:10 (L:D), the developmental duration was the shortest and the number of emerged adults was the highest. Adult life span was the longest under a 12:12 (L:D) photoperiod. During the adult stage, supplementary nutrition, such as sucrose, fructose, honey, and glucose, increased the life span of *S. caerulea*. The life span of *S. caerulea* was longer when provided with a supplementary diet of sucrose or honey, compared to other tested diets. The results suggested that the most suitable conditions for *S. caerulea’s* population growth were the following: 30 to 33 °C, with 12 to 14 h of daylight, and the provision of sucrose or honey as supplemental diet for the adults.

## 1. Introduction

Environmental factors, such as temperature, humidity, and photoperiod, affect the growth, reproduction, and behavioral activities of parasitoids and other insects and ultimately lead to population changes [1]. Temperature directly affects parasitoid development, fecundity, and survival [2,3]. To complete their development, parasitoids must accumulate a certain amount of heat above a lower developmental threshold temperature, which is called their effective accumulated temperature. Developmental threshold temperature, effective accumulated temperature, and suitable temperature are important factors determining the suitability of a geographical region for parasitoids. When environmental temperatures exceed a parasitoid’s lower developmental threshold temperature, development begins. At suitable temperatures, the parasitoids grow and reproduce effectively. When experiencing unsuitable high or low temperatures, however, the growth and reproduction of parasitoids may be inhibited [4,5]. Additionally, changes in available light can cause parasitoids to adjust their metabolism and diapause [6]. Therefore, photoperiod plays an important role in regulating the growth and reproduction of parasitoids [7,8]. Changes in photoperiod can impact the parasitism rate, emergence rate, life span and sex ratio of parasitoids [9,10]. Furthermore, parasitoid adults primarily obtain nutrients by feeding on nectar, plant exudates (honeydew), host tissues or hemolymphs [11,12,13]. Supplementary nutrition for parasitoid adults usually consists of energy-providing carbohydrates, which affect ovarian development and adult life span [14,15]. As the structure and function of carbohydrates differ, various nutrient supplements can affect parasitoids differently.

*Parasaissetia nigra* Nietner (Hemiptera: Coccidae) is an invasive species of scale insect that is native to Africa, but currently has a broad global distribution [16]. *Parasaissetia nigra* Nietner is highly polyphagous and has been documented to feed on plants belonging to more than 92 families [17]. *Parasaissetia nigra* Nietner is an important rubber trees pest and has severely affected the rubber industry in tropical areas of China, such as Yunnan and Hainan [18]. Fortunately, there are several biological controls for *P. nigra*, such as *Coccophagus ceroplastae* Howard, *Metaphycus parasaissetiae* Zhang and Huang, *Coccophagus japonicus* Compere, and *Diversinervus elegans* Silvestri [19,20,21,22]. Normally, the optimum temperature range for the reproduction and development of these parasitoids is between 20 and 30 °C [23,24,25]. However, the summer temperature in the tropical areas where rubber trees are planted often exceeds 30 °C, which limits the establishment of field populations of these parasitoids.

Unlike the other major parasitoids of *P. nigra*, the parasitoid *Scutellista caerulea* Fonscolombe (Hymenoptera: Pteromalidae) can provide an effective control for *P. nigra* in high-temperature environments. *Scutellista caerulea* Fonscolombe is an egg parasitoid of multiple species of scale insects, including *Ceroplastes rusci* Linnaeus, *Saissetia oleae* Olivier, *Coccus hesperidum* Linnaeus, and *Ceroplastes floridensis* Comstock [26,27,28]. Larvae of *S. caerulea* are ectoparasitic and feed on host eggs under its abdomen [29]. A survey conducted by Awamelah et al. (2009) in Jordan found that the *S. caerulea* was highly abundant from September to October, and that its population density significantly decreased from November to December [30]. Badary and Abd-Rabou (2011) investigated the parasitism rate of *S. caerulea* at eight locations in Egypt and found that its parasitism rates were highest from September to October [28]. Additionally, *S. caerulea* was recently found in parasitized *P. nigra* collected in a rubber tree nursery in Hainan, China, with more than 60% of parasitism rate in the field [31].

Therefore, the indoor rearing of *S. caerulea* may help control *P. nigra* in the field. However, the necessary conditions for the successful indoor rearing of *S. caerulea*, such as temperature, photoperiod, and supplementary nutrition, are still unclear. This study aimed to clarify the suitable temperature, photoperiod, and supplementary nutrition for establishing an indoor population of *S. caerulea*. This study provides an empirical basis for rearing *S. caerulea*.

## 2. Materials and Methods

### 2.1. Insects

*Parasaissetia nigra* Nietner adults were collected from rubber trees in an experimental station at the Yunnan Institute of Tropical Crops in 2004 (20.05° N, 102.72° E). The rearing of these insects was performed in a laboratory at 25 to 27 °C and 70 to 90% relative humidity. All insects were fed with pumpkin fruits.

*Scutellista caerulea* Fonscolombe pupae were collected from a rubber plantation at Xinjin Farm in Qiongzhong, Hainan Province (19.03° N, 109.84° E). After emergence, the insects were reared with *P. nigra* to generate a population sufficient for the experiment.

### 2.2. Development and Reproduction of S. caerulea at Different Temperatures

To determine the effects of temperature on the development and reproduction of *S. caerulea*, several temperature experiments were conducted. All experiments were performed in artificial climate chambers (MGC-350HP-2, Shanghai Yiheng Scientific Instrument Limited company, Shanghai, China) (70% RH and 12:12 (L:D) photoperiod). The tested temperatures were 18, 21, 24, 27, 30, 33, and 36 °C. The tested hosts were adult *P. nigra* with a hard shell and black body, which laid eggs for 1 to 2 days. Four replicates were conducted for each treatment.

To assess how temperature influenced the oviposition and parasitism rates, two newly emerged virgin females (<5 h old) were paired with two newly emerged virgin males (<5 h old) for one day. The two females were then put together in a petri dish (Φ 9.0 cm), and 30 *P. nigra* were added to the dish for parasitization. Afterward, the dishes were kept in artificial climate chambers at different temperatures. After 24 h, the scales were dissected under a stereomicroscope (JSZ8, Nanjing Jiangnan Yongxin Optical Limited company, Nanjing, China) to count the number of eggs laid by the females and calculate the parasitism rates. The parasitism rates of *S. caerulea* were calculated as percentages of parasitized *P. nigra* over the total tested *P. nigra*.

To test how temperature affected the developmental duration and number of emerged adults of *S. caerulea*, 30 *P. nigra*, reared on pumpkin fruits, were covered with a transparent plastic cup (Φ 7.5 cm and a height of 8.5 cm). The edge of the cup was glued with a circular sponge, and the bottom of the cup was opened with a 1.1 cm diameter hole. Two newly emerged virgin *S. caerulea* females were paired with two newly emerged virgin males for one day. The two females were then released into the plastic cup. After 24 h at 27 °C, the *S. caerulea* adults were removed and the *P. nigra* were reared at different temperatures. The developmental duration and the number of emerged *S. caerulea* adults were recorded daily, and the developmental threshold temperature and effective accumulated temperature were calculated.

The development rate (v) of *S. caerulea* refers to the development duration (D) in days, v=1/D. The developmental threshold temperature and effective accumulated temperature were calculated using both the linear regression method and the optimum seeking method to ensure a rather complete view [32,33].

For the linear regression method, the developmental threshold temperature was calculated using the following formula: $C=(\;\sum^{\;}v^{2}\sum^{\;}T-\sum^{\;}v\sum^{\;}v\;T)/(n\sum^{\;}v^{2}-(\sum^{\;}{v)}^{2})$, where C is the developmental threshold temperature, v is the development rate, n is the number of temperature treatments, and T is the temperature in the experiment. The effective accumulated temperature was calculated using the following formula: $K=(n\sum^{\;}v\;T-\sum^{\;}v\sum^{\;}T)/(n\sum^{\;}v^{2}-(\sum^{\;}{v)}^{2})$, where K is the effective accumulated temperature, v is the development rate, n is the number of temperature treatments, and T is the temperature in the experiment.

For the optimum seeking method, the developmental threshold temperature was calculated using the following formula: $C=(\sum_{i=1}^{n}D_{i}^{2}T-D\sum_{i=1}^{n}D_{i}T)/(\sum_{i=1}^{n}D_{i}^{2}{-\mathrm{nD}}^{2})$, where C is the developmental threshold temperature, n is the number of temperature treatments, T is the temperature in the experiment, and D_i_ is the development period at this temperature. The effective accumulated temperature was calculated according to the following formula: $K=\frac{1}{n}\sum_{i=1}^{n}D_{i}{(T}_{i}-C)$, where C is the developmental threshold temperature, K is the effective accumulated temperature when the developmental threshold temperature is C, n is the number of temperature treatments, T is the temperature in the experiment, and D_i_ is the development period at this temperature.

To determine how temperature affected the female ratio of *S. caerulea*, 20 newly emerged virgin females, 20 newly emerged virgin males, and 200 *P. nigra* were put in cages (30 cm × 30 cm × 30 cm), which were kept at different temperatures. After 24 h, the *P. nigra* were taken out of the cages and continued to feed at different temperatures until the parasitoid adult emergence, and then the number and sex of the emerged *S. caerulea* was determined. The female ratio of *S. caerulea* was calculated as the percentage of emerged females of the total emerged *S. caerulea*.

To assess how temperature affected the life span of the adult *S. caerulea*, experiments were conducted in three test tubes (Φ 1.2 cm and a length of 6.0 cm, with mesh lids). Five newly emerged virgin females and five newly emerged virgin males were place in each tube, respectively. Absorbent cotton, dipped with 15% sucrose water, was placed on each tube’s wall for nutrition, and the test tubes were assigned to different temperature treatments. The tubes were checked daily between 8:00 a.m. and 10:00 a.m. local time to count the number of dead females and males.

### 2.3. Development and Reproduction of S. caerulea at Different Photoperiods

To determine the effects of photoperiod on the development and reproduction of *S. caerulea*, several photoperiod experiments were conducted. All experiments were performed in artificial climate chambers (70% RH and 27 °C). To assess the number of eggs laid and the parasitism rates of *S. caerulea* at different photoperiods, the tested photoperiods were 0:24, 2:22, 4:20, 6:18, 8:16, 10:14, 12:12, 14:10, 16:8, 18:6, 20:4, 22:2, and 24:0 (L:D). For other experiments, the tested photoperiods were 8:16, 10:14, 12:12, 14:10, and 16:8 (L:D). Tested hosts were adult *P. nigra* with a hard shell and black body, which laid eggs for 1 to 2 days. Four replicates were conducted for each treatment.

To assess how photoperiod influenced oviposition and parasitism rates, two mated females were then placed in a petri dish, and 30 *P. nigra* were added to the dish for parasitization. The dishes were then maintained at different photoperiods. After 24 h, the scales were dissected under a stereomicroscope to count the number of eggs laid by the females and to calculate the parasitism rates.

To test how photoperiod affected the developmental duration and the number of emerged adults of *S. caerulea*, 30 *P. nigra* reared on pumpkin were covered with a transparent plastic cup. Two mated females were then released into the plastic cup. After 24 h under a 12:12 (L:D) photoperiod, the *S. caerulea* adults were removed, and the *P. nigra* were reared under different photoperiods. The developmental duration and number of emerged adults of *S. caerulea* were recorded daily.

To determine how photoperiod affected the female ratio of *S. caerulea*, 20 newly emerged virgin females, 20 newly emerged virgin males, and 200 *P. nigra* were placed in cages (30 cm × 30 cm × 30 cm) maintained under different photoperiods. After 24 h, the *P. nigra* were taken out of the cages and continued to feed under different photoperiods until the parasitoid adult emergence, and then the number and gender of *S. caerulea* that emerged from the *P. nigra* were recorded.

To assess how photoperiod affected the life span of the adult *S. caerulea*, experiments were conducted in three test tubes. Five newly emerged virgin females and five newly emerged virgin males were placed in each tube, respectively. Absorbent cotton, dipped with 15% sucrose water, was placed on each tube’s wall for nutrition, and the test tubes were assigned to different photoperiod treatments. The tubes were checked daily between 8:00 a.m. and 10:00 a.m. local time to count the number of dead females and males.

### 2.4. Development and Reproduction of S. caerulea under Different Supplementary Nutrition

To determine how supplementary nutrition affected the number of eggs laid by females, a mated female was then placed in a petri dish. Absorbent cotton was dipped with either a 20% sucrose, melezitose, fructose, honey, glucose, or trehalose solution and placed on the dish as supplementary nutrition. Water and no nutrition were used as the controls. Fifteen *P. nigra* (adults with a hard shell and black body, which laid eggs for 1 to 2 days) were placed in the dish for *S. caerulea* parasitization. The dishes were kept in artificial climate chambers (27 °C, 70% RH, and 12:12 (L:D) photoperiod). Every 24 h during the egg-laying period, scales were dissected under a stereomicroscope to determine the number of eggs laid and the number of scales parasitized. Fifteen new scales were replaced daily. Each treatment was repeated four times.

To determine how supplementary nutrition affected the life span of adult *S. caerulea*, three test tubes were used. Five newly emerged virgin females and five newly emerged virgin males were placed in each tube. Absorbent cotton was dipped into 20% sucrose, melezitose, fructose, honey, glucose or trehalose solution, respectively, and placed on the tube walls as supplementary nutrition. Water and no nutrition were used as the controls. The test tubes were kept in artificial climate chambers (27 °C, 70% RH and 12:12 (L:D) photoperiod). The tubes were checked daily between 8:00 a.m. and 10:00 a.m. local time to count the number of dead females and males. Each treatment was repeated four times.

### 2.5. Data Analysis

Data in the figures are stated as means ± standard errors. Parasitism rates and female ratios were analyzed by logistic regression, and developmental duration was analyzed using two-way ANOVA, while other data were analyzed using one-way ANOVA. A *p* value of <0.05 was considered to be statistically significant. All the data were analyzed using SPSS 23.0 for Windows (http://www.spss.com). The figures and tables were prepared using Microsoft Excel 2016.

## 3. Results

### 3.1. Effects of Temperature on the Development and Reproduction of S. caerulea

We observed that *S. caerulea* could lay eggs at 18 to 36 °C, but it could only complete generational development at 21 to 33 °C. Therefore, the data in this article on developmental duration, the number of emerged adults, female ratio, and adult life span, were only statistically based on the range of 21 to 33 °C. On the other hand, the analysis focusing on the number of eggs laid by females and the parasitism rate (based on the standard of laying eggs) were based on data collected at 18 to 36 °C.

Temperature significantly influenced the number of eggs laid by *S. caerulea* (one-way ANOVA, *F*
_(6, 21)_ = 26.808, *p* < 0.001). The mean number of eggs laid by *S. caerulea* at 33 °C was significantly greater than the number of eggs laid at the other temperature treatments (Figure 1a). Temperature also significantly influenced the parasitism rates of *S. caerulea* (Logistic Regression, *Wale* = 59.401, *p* < 0.001). The parasitism rate was significantly greater at 33 °C (77.78 ± 4.01%) than that observed at other temperatures, except for the rate at 30 °C (Table 1 and Figure 1b).

Temperature significantly influenced the developmental duration of *S. caerulea* (two-way ANOVA, *F*
_(4, 30)_ = 3534.106, *p* < 0.001). From 21 to 33 °C, the development of *S. caerulea* increased with increase of temperature. The developmental durations of females and males were the longest at 21 °C. These were significantly longer than developmental durations at other temperatures. The effects of temperature on developmental duration of *S. caerulea* males and females were different. At 21 and 24 °C, the development of males was faster than that of females (two-way ANOVA, *F*
_(1, 30)_ = 40.529, *p* < 0.001). The interaction of temperature and sex significantly influenced the developmental duration of *S. caerulea*, (two-way ANOVA, *F*
_(4, 30)_ = 7.194, *p* < 0.001) (Figure 1c). The number of emerged adults significantly increased when the temperature increased (one-way ANOVA, *F*
_(4, 15)_ = 39.230, *p* < 0.001). The number of emerged adults was significantly higher at 33 °C (18.3 ± 1.5) than the number of emerged adults observed at other temperatures, except for at 30 °C (Figure 1d).

Temperature significantly influenced the female ratio of *S. caerulea* when temperature was in the range of 21 to 33 °C (Logistic Regression, *Wale* = 7.817, *p* = 0.005). The female ratio was highest at 24 °C. However, there was no significant difference amongst different temperatures in the range of 21 to 30 °C (Table 2 and Figure 1e). Temperature significantly influenced the life span of adult *S. caerulea* at 21 to 33 °C (one-way ANOVA, *F* _(4, 15)_ = 53.198, *p* < 0.001). The longest lifespan was observed at 24 °C, even though there was no significant difference amongst different temperatures in the range of 21 to 27 °C (Figure 1f).

Based on linear regression methods, the developmental threshold temperatures of the female and male *S. caerulea* were 14.18 °C and 13.93 °C, respectively. The effective accumulated degree-days for females and males were 335.86 and 330.61, respectively. Using the optimum seeking method, the developmental threshold temperatures of the female and male *S. caerulea* were 15.21 °C and 14.97 °C, respectively. The effective accumulated degree-days for females and males were 307.00 and 302.71, respectively (Table 3).

### 3.2. Effects of Photoperiod on the Development and Reproduction of S. caerulea

Photoperiod significantly influenced the number of eggs laid by *S. caerulea* (one-way ANOVA, *F*
_(12, 39)_ = 105.334, *p* < 0.001). When the daily light time was 0 h females did not lay eggs. When the daily light time increased, the number of eggs laid by females significantly increased. When the light times were 24 h and 22 h, the number of eggs was 43.3 ± 4.4 and 43.0 ± 3.5, respectively. There was no significant difference between them, but both were significantly greater than the other treatments (Figure 2a). Photoperiod also significantly influenced the parasitism rate of *S. caerulea* (Logistic Regression, *Wale* = 146.124, *p* < 0.001). When the daily light time was 0 h, the parasitism rate was zero. As the light time increased, the parasitism rates of *S. caerulea* increased. When the light time reached 14 h or more in one day, there was no significant difference in the parasitism rates among different daily light times. When the light time was 24 h, the parasitism rate was the greatest (89.17 ± 5.7%) (Table 4 and Figure 2b).

Photoperiod significantly influenced the developmental duration of *S. caerulea* (two-way ANOVA, *F*
_(4, 30)_ = 24.338, *p* < 0.001). *S. caerulea* development was fastest under a 14:10 (L:D) photoperiod. The developmental durations of females and males were 22.3 ± 0.2 d and 20.3 ± 0.2 d, respectively, and these times were significantly shorter than those in the other treatments. The effects of photoperiod on developmental duration of *S. caerulea* males and females were different (two-way ANOVA, *F*
_(1, 30)_ = 57.970, *p* < 0.001), but the interaction of photoperiod and sex did not significantly affect the developmental duration of *S. caerulea* (two-way ANOVA, *F*
_(4, 30)_ = 1.181, *p* = 0.3495) (Figure 2c). Photoperiod significantly influenced the number of emerged adults of *S. caerulea* (one-way ANOVA, *F*
_(4, 15)_ = 18.042, *p* < 0.001). When the photoperiod was 14:10 (L:D), the number of emerged adults was greatest (17.0 ± 0.6), and this was significantly different from the other treatments, with the exception of the 16:8 (L:D) photoperiod (Figure 2d).

When the photoperiods were 8:16, 10:14, 12:12, 14:10, and 16:8 (L:D), the female ratios were not significantly different among the treatments (Logistic Regression, *Wale* = 2.298, *p* = 0.13) (Table 5 and Figure 2e). Photoperiod significantly influenced the life span of adult *S. caerulea* (one-way ANOVA, *F*
_(4, 15)_ = 12.909, *p* < 0.001). When the photoperiod was 12:12 (L:D) adults had the longest life span (33.1 ± 0.7 d) (Figure 2f).

### 3.3. Effects of Supplementary Nutrition on the Development and Reproduction of S. caerulea

Supplementary nutrients did not affect the number of eggs laid by *S. caerulea* (one-way ANOVA, *F*
_(7, 24)_ = 1.458, *p* = 0.231), but the number of eggs were different when glucose and trehalose were used as supplementary nutrients (Figure 3a). Supplementary nutrients did not affect the number of *P. nigra* parasitized by females either (one-way ANOVA, *F*
_(7, 24)_ = 1.666, *p* = 0.167), but the number of *P. nigra* parasitized by females were higher when using glucose or honey, compared to trehalose, as supplementary nutrients (Figure 3b).

Supplementary nutrition significantly affected the life span of adult *S. caerulea* (two-way ANOVA, *F*
_(7, 48)_ = 218.442, *p* < 0.001). When *S. caerulea* adults were provided with supplementary sucrose, the life spans of adult females (33.0 ± 1.3 d) and males (33.2 ± 0.6 d) were the longest. The effects of supplementary nutrients on life spans of *S. caerulea* males and females were different (two-way ANOVA, *F*
_(1, 30)_ = 57.970, *p* = 0.026). When supplemented with fructose, females lived significantly longer than males. The interaction of supplementary nutrition and sex also significantly influenced the life span of adult *S. caerulea* (two-way ANOVA, *F*
_(4, 30)_ = 1.181, *p* = 0.014). (Figure 3c).

## 4. Discussion

The temperature of the environment is one of the most important factors affecting the development and reproduction of parasitoids [34,35]. Our study demonstrated that 30 to 33 °C is the optimal temperature for the population growth of *S. caerulea*. The number of eggs laid was the most, the parasitism rate of *S. caerulea* was the highest, the developmental period was the fastest, and the number of emerged adults was the most at 33 °C. On the other hand, the female ratio was lower and life span was shorter at 33 °C. However, there was no significant difference in the female ratio when the temperature was 30 °C or lower. Additionally, the adult life span should be less important for indoor proliferation than parasitism rate, developmental duration, and the number of emerged adults of this parasitoid, as parasitism of *S. caerulea* is monoparasitism. The effect of temperature on developmental duration of female and male parasitoids has been found to be different [36]. Our study found that female developmental durations of *S. caerulea* were typically longer than that of males under 21 and 24 °C. In our study, the female life spans were estimated using females not laying eggs. However, in the field, the females always lay eggs, so future studies should examine the impact of oviposition on life span. At 33 °C the female ratio of *S. caerulea* was significantly lower than that at other tested temperatures, demonstrating that there were more males at high temperatures. This might indicate that *S. caerulea* lays fewer female eggs, or the survival rate of females is lower than males, when the temperature is higher [37]. Further study is required to better understand the mechanism of this phenomenon. Under 18 °C (low-temperature condition) and 36 °C (high-temperature condition), female *S. caerulea* could still lay eggs, but the parasitoids could not complete their development. The reason for this might be direct or indirect. The parasitoid eggs might die at too high temperatures. On the other hand, their hosts might die at too high temperatures, indirectly resulting in the parasitoid eggs failing to develop. For example, *P. nigra* could not complete development, when the temperature was higher than 35 °C [38]. If *S. caerulea* parasitizes a high-temperature-resistant host, it is unclear whether it can complete development even at higher temperature, so this remains to be further studied.

Another important factor for parasitoid development and reproduction is photoperiod [39,40]. This study demonstrated that the oviposition of *S. caerulea* was positively correlated with the length of the daily light time. The parasitism rates of *S. caerulea* increased with the increase of light time when the light time was less than 14 h/d. However, when the light time exceeded 14 h, the parasitism rates remained stable. Considering that when the light time exceeded 14 h the parasitism rate was above 80%, these was little room to grow, so, consequently, we could infer that the parasitism rate reached a plateau at the regimen of 14 h of light. Although at 14 h/d, the number of eggs laid was not the highest, this parasitoid wasp is a single parasitoid, so a host can only feed one parasitoid offspring, and, therefore, the parasitism rate is more important than the number of eggs laid. Therefore, a 14 h daily light time was beneficial to the reproduction of *S. caerulea*. In addition, studies have shown that photoperiod can affect the developmental duration of parasitoids [41,42]. When the daily light time was between 8 and 16 h, the developmental duration, number of emerged adults, and adult life spans of *S. caerulea* differed at different photoperiods, but their entire life history was unaffected. For *S. caerulea*, development was fastest, and number of emerged adults was largest, under a 14:10 (L:D) photoperiod, which indicated that this photoperiod was the most beneficial to the development of their larvae. We also found that a 12:12 (L:D) photoperiod was the most beneficial to the survival of *S. caerule* adults. The adult life span decreased when the daily light time was shortened or prolonged. It has also been reported that photoperiod affects the sex ratio of parasitoids. For example, some studies have found the female ratio of parasitoids decreased when the daily light time increased [9], while some have found the opposite phenomenon, where a long-day photoperiod could promote more female offspring [10]. However, our study found that photoperiod had no significant effect on the sex ratio of *S. caerulea*. Therefore, based on the above results, a 12 to 14 h daily light time was a suitable photoperiod for raising *S. caerulea* indoors.

After emergence, most adult parasitoids continue to feed on supplementary nutrition. Supplementary nutrition can increase the number of eggs laid, the parasitism rate, and the life span of parasitoids to benefit their reproduction [43,44]. For example, a previous study by Wu et Chen reported that supplementing the diet with honey, glucose, sucrose, and fructose could increase the number of eggs laid by *Telenomus theophilae* [45]. We found that *S. caerulea* adults fed on carbohydrate sources such as sucrose, melezitose, fructose, honey, glucose, and trehalose. In addition, there are many previous studies confirming that supplementary nutrition significantly prolongs the life span of parasitoid wasps [46,47,48]. Our study also found that supplementary nutrition could extend the adult life span of *S. caerulea*. When sucrose and honey were used as supplementary nutrition, the life spans were longer than those in the other treatments. This may be because honey contains not only sugar, but also vitamins and proteins, which is closer to the food of parasitoid adults in the wild condition [49]. In conclusion, according to our experimental results, when rearing *S. caerulea* indoors, it is better to choose sucrose or honey for supplementary nutrition.

The data obtained from our experiments are valuable for the mass-rearing of *S. caerulea* indoors, but there are also many deficiencies. We reared *S. caerulea* indoors with the ultimate goal of releasing *S. caerulea* into the field to control the rubber tree pest *P. nigra*, but our results were not entirely applicable to the field. In fact, in the field, the temperature is not in a constant state (as it is in the indoor state) and is always fluctuating [50,51]. The same is true for the daily light time and intensity with the weather or seasonal changes. In addition, adults of most parasitoid species feed on floral nectar and plant exudates in the field. The nutritional content of these foods is more complex. Although they are mainly composed of glucose, fructose and sucrose, they also contain small amounts of other sugars, lipids, amino acids and even protein components [52,53]. Moreover, our experiment did not take into account the effect of humidity. Humidity is another important factor in parasitoid reproduction [54]. Therefore, we need to broaden our experimental design in the future, not only aiming to rear *S. caerulea* indoors, but also simulating the development and reproduction of *S. caerulea* in the field, for better control of target pests.

## 5. Conclusions

In conclusion, the present study demonstrated the best conditions for rearing *S. caerulea*, which should be reared at 30 to 33 °C, with 12 to 14 h of daily light time, and using sucrose or honey as the supplementary nutrition for the adults. To reduce costs for practical propagation, sucrose would be more suitable as the supplementary nutrition. Future studies need to examine the role of biotic and abiotic factors on parasitoid biocontrol under field conditions.

## Figures and Tables

**Figure 1 insects-14-00082-f001:**
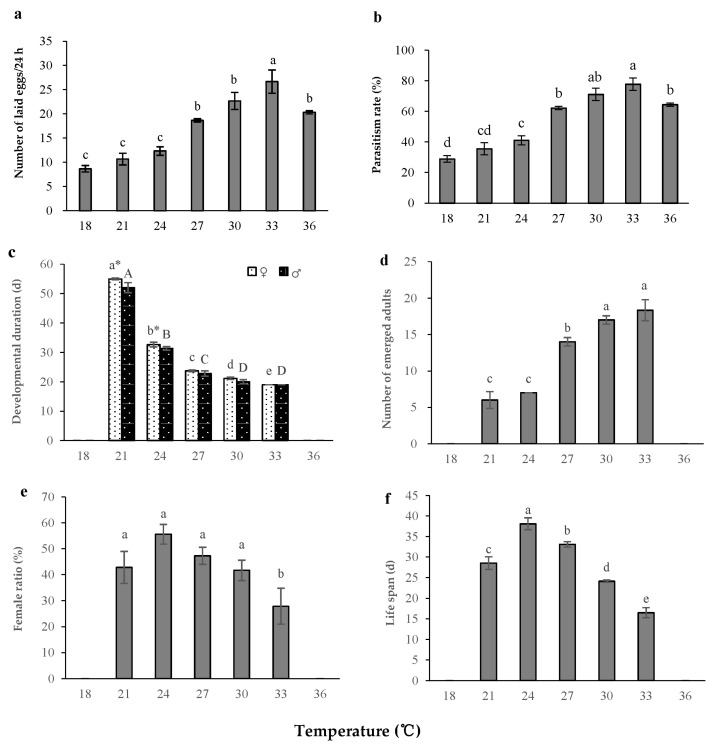
Effects of temperature on the development and reproduction of *S. caerulea*. (**a**) Number of eggs laid (the eggs were laid by two females during 24 h). (**b**) Parasitism rates. (**c**) Developmental duration. (**d**) Number of emerged adults. (**e**) Female ratio. (**f**) Adult life span. means ± standard errors, the different letters indicate significant differences amongst the temperatures (female, capital letters; male, lowercase letters) (*p* < 0.05), and * indicates significant differences between females and males at the same temperatures (*p* < 0.05).

**Figure 2 insects-14-00082-f002:**
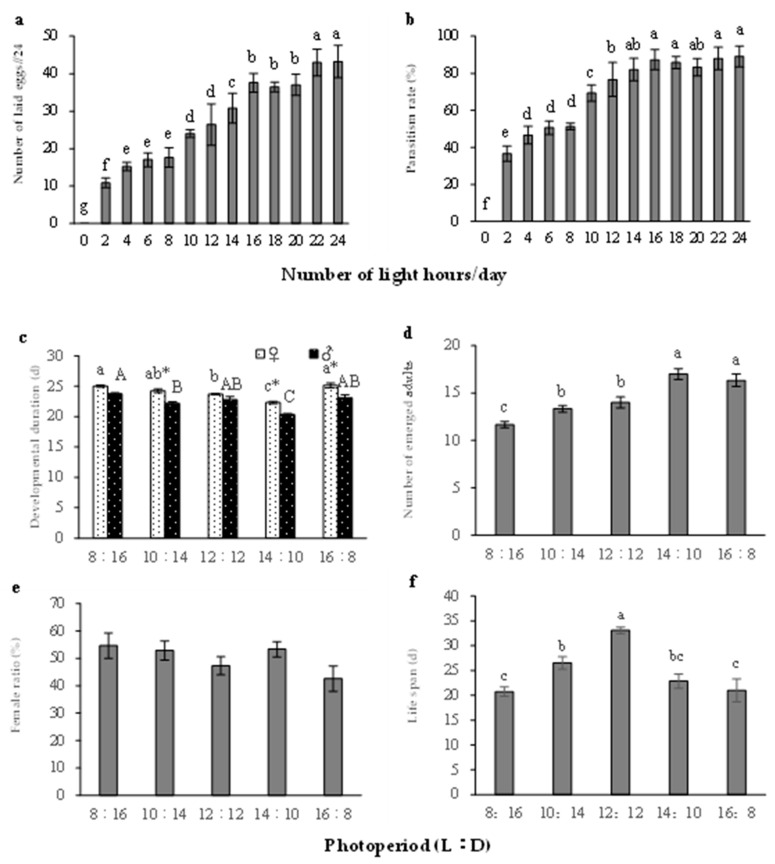
Effects of photoperiod on the development and reproduction of *S. caerulea*. (**a**) Number of eggs laid (the eggs were laid by two females during 24 h). (**b**) Parasitism rates. (**c**) Developmental duration. (**d**) Number of emerged adults. (**e**) Female ratio. (**f**) Adult life span. means ± standard errors, the different letters indicate significant differences amongst the photoperiods (female, capital letters; male, lowercase letters) (*p* < 0.05), and * indicates significant differences between females and males at the same photoperiods (*p* < 0.05).

**Figure 3 insects-14-00082-f003:**
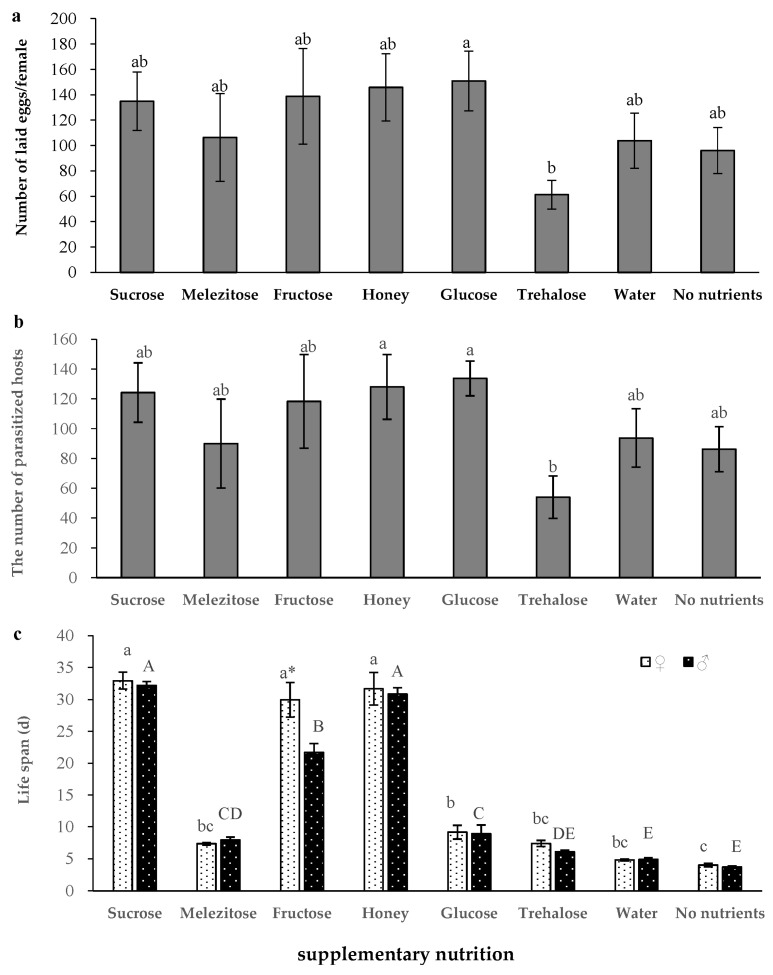
Effects of supplementary nutrition on the development and reproduction of *S. caerulea*. (**a**) Number of eggs laid (the eggs were laid by each female throughout their entire lifespan). (**b**) Number of hosts parasitized. (**c**) Adult life span. means ± standard errors, the different letters indicate significant differences amongst the supplementary nutrition (female, capital letters; male, lowercase letters) (*p* < 0.05), and * indicates significant differences between females and males having the same supplementary nutrition (*p* < 0.05).

**Table 1 insects-14-00082-t001:** Logistic regression analysis of the effect of temperature on *S. caerulea*’s parasitism.

Variable	*B*	*Wald*	*OR* (95% *CI*)	*p-Value*
temperature	0.113	59.401	1.120 (1.088–1.152)	0.000

**Table 2 insects-14-00082-t002:** Logistic regression analysis of the effect of temperature on *S. caerulea*’s female ratio.

Variable	*B*	*Wald*	*OR* (95% *CI*)	*p-Value*
temperature	−0.60	7.817	0.941 (0.902–0.982)	0.005

**Table 3 insects-14-00082-t003:** Lower developmental threshold temperature and effective accumulative temperature of *S. caerulea*.

Method	Lower Developmental Threshold Temperature (°C)		Effective Accumulative Temperature (Degree-Days)	
	Female	Male	Female	Male
Linear regression method	14.18	13.93	335.86	330.61
Optimum seeking method	15.21	14.97	307.00	302.71

Note: degree-day is the unit of the effective accumulated temperature.

**Table 4 insects-14-00082-t004:** Logistic regression analysis of the effect of photoperiod on *S. caerulea*’s parasitism.

Variable	*B*	*Wald*	*OR* (95% *CI*)	*p-Value*
temperature	0.265	146.124	0.908 (0.802–1.029)	0.000

**Table 5 insects-14-00082-t005:** Logistic regression analysis of the effect of photoperiod on *S. caerulea*’s female ratio.

Variable	*B*	*Wald*	*OR* (95% *CI*)	*p-Value*
temperature	−0.96	2.298	1.120 (1.088–1.152)	0.130

## Data Availability

The data presented in this study are available upon request from the corresponding author. The data are not publicly available due to the uncompleted subject.

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
