# Peer review of "Influence of Temperature, Photoperiod, and Supplementary Nutrition on the Development and Reproduction of *Scutellista caerulea* Fonscolombe (Hymenoptera: Pteromalidae)"

_insects, 2023, doi:10.3390/insects14010082_

Round 1

Reviewer 1 Report

The authors conducted a very simple laboratory study, but a lot of data were obtained that can be very important for mass rearing of a potential biocontrol agent. Thus, the paper deserves to be published, although a number of minor corrections should be made (see below). In addition, although I am also not a native speaker, I think that the manuscript would also require intensive editing of English grammar.

Line 29: replace “from 18-33°C” with “from 18 to 33°C” (the same in lines 30 and 32).

Lines 50 and 54: replace “minimum developmental threshold” with “lower developmental threshold”

Line 56: replace “periods unsuitably” with “periods of unsuitably”

Lines 58-61: I would add that photoperiod can play a significant role in the induction of insect diapause (that is one of the most ecologically important effects of day length).

Line 70: P. nigra should be in Italics (the same in line 72). Besides generic Latin names in the beginning of a sentence should not be abbreviated.

Line 80: Generic Latin names in the beginning of a sentence should not be abbreviated (replace “S. caerulea” with “Scutellista caerulea”).

Lines 93-95: an incomplete sentence.

Lines 100 and 104: Again, generic Latin names in the beginning of a sentence should not be abbreviated.

Line 105: I guess, you mean not “feathering” but rather “emergence”?

Line 117: replace “Dishes” with “dishes”.

Lines 174 and 194: If you indicate the experimental checking time, not local astronomic time, but time relatively to the light-on (or light-off) of the artificial photoperiod should be indicated.

Line 235: replace “from 21-33°C” with “from 21 to 33°C”

Line 239: replace “the development of male faster than female” with “the development of males was faster than that of females”.

Line 247: replace “At 24°C was the highest” with “At 24°C female ratio was the highest”

Line 316, Table 1: replace “The development threshold” with “The lower development threshold” (do the same in Table 1).

Figures 1-3, Y-axes: Delete all zeros after the decimal point.

Figures 1-3, a, Y-axes: replace “The number of eggs (grain)” with “The number of laid eggs”.

Figures 1-3, d, Y-axes: replace “The number of emergence (head)” with “The number of emerged adults”

Figure 3, b, Y-axes: Delete “(head)”.

Lines 378-379: As clearly seen in Fig. 3, both the number of laid eggs and the parasitism rate with feeding on sucrose, fructose, honey, or glucose were NOT significantly different from those in unfed controls. Thus, this conclusion was NOT supported by the data. Please, note that in the Results (lines 296-299) it is also stated that supplementary food has no impact on the number of eggs laid and the number of P. nigra parasitized.

Line 391: replace “Formosa” with “formosa”.

Reviewer 2 Report

General comments

Overview: This is a very simple and basic biological study to determine the effects of some biotic and abiotic factors on a few life-history traits of a parasitoid (Scutellista caerulea) that has potential to biocontrol an important pest (Parasaissetia nigra) of the rubber tree. While I do appreciate the efforts of the authors to explore these questions, I am afraid I have way too many motives to recommend this manuscript to be rejected for publication. Please, find below the many reasons why.

English language: I think this manuscript would strongly benefit from an English review by a native English speaker.

Discussion Section: This section is too short and not well-written. Most of the text refers to results that were already presented in the Results section. There is virtually no discussion at all. Sometimes the authors make statements that are not supported by their data at all or that are even contradicted by their data. The authors often compare their results for S. caerulea with those for other parasitoid species, especially those from other hymenopteran families. Also, the authors make very obvious affirmations such as "The number of emerged adults was largest under 14:10 (L:D) photoperiod, which indicated that the survival rate of S. caerulea was higher under this photoperiod." Seriously? That is not scientific discussion. 

Other general comments:

Lines 142-175, 188-195: Most of these methods are identical to those ones described in the lines 108-140, so there is no need to repeat everything. Just write a sentence or two stating that the methods are similar to those previously mentioned.

Figures 1a, 2a: Replace "The number of eggs (grain)" with "Number of eggs/female/24h" and adjust the number of females and/or hours if that is the case.

Figure 1cdef: Where are the results for 18 and 36ºC? If there was no emergence at these temperatures, that information must be within the Results section and at Figure 1cdef.

Figures 1d, 2d: Replace "The number of emergence (head)" with "Number of emerged adults/female/24h" and adjust the number of females and/or hours if that is the case.

Figures 1f and 2f: Since there is no significant effect of sex, there is no reason to present separated columns for males and females. Also, all those pairs of letters are confusing since there is no significant difference between males and females. There is no reason to present a pair of letters for each temperature/photoperiod, there should be only one letter for each temperature/photoperiod.

Table 1: Replace "ºC-d" with "degree-days". Use the "Note" to explain that those degree-days are in ºC.

Figure 2ab: Include the title of the X axis (Number of light hours/day).

Figure 2c: Photoperiods 8:16 and 12:12 are missing asterisks to indicate significant differences between males and females, as stated at lines 278-282.

Figure 2e: Since there is no significant difference amongst temperatures, I suggest deleting the letters "a".

Figures 3a: Replace "The number of eggs (grain)" with "Number of eggs/female".

Specific comments and suggestions

Line 20: Replace "Homoptera" with "Hemiptera".

Line 25: Replace "Scutellista caerulea Fonscolombe" with "S. caerulea".

Lines 29, 30 and 32: Replace "From" with "At".

Lines 42-43: All keywords are already in the title. I suggest replacing them with keywords that are not part of the title. 

Line 53: Geographical "region"?

Line 56: "periods [of] unsuitably…"

Line 62: Replace "honeydews" with "honeydew".

Line 67: Replace "Homoptera" with "Hemiptera".

Lines 70, 72: Use italic for "P. nigra".

Line 72: I think the "C" in "Coccophagus" needs to be italicized; Replace "Howard." with "Howard,".

Lines 82-83: If the larvae of S. caerulea feed on various eggs of P. nigra, is S. caerulea still considered a parasitoid? Wouldn't it be a predator instead? By definition, a parasitoid needs a single host to complete development. Please, see the definition of parasitoids by Mills (2009) attached at the end of this document.

Line 84: Delete "and".

Line 93: Replace "a population" with "an indoor population" or "a laboratory population".

Lines 93-95: Delete "By examining the effects of temperature, photoperiod and supplementary nutrition on the development and reproduction of S. caerulea."

Line 95: Replace "theoretical" with "empirical".

Lines 95-97: Delete "and the field application" and "and contributes to the environmentally friendly and sustainable control of P. nigra".

Line 101: Do you really mean "mass rearing" (millions of individuals per week) or just a small lab colony for experiments?

Line 105: What does "feathering" mean here? Do you mean adult emergence?

Lines 107, 141, 176, 227, 263, 295: "S. caerulea" should be written with a font that differs from the rest of the text.

Line 116: The females were put together or individually into petri dishes?

Line 117: Replace "Afterward, Dishes" with "Afterwards, dishes".

Lines 123-124, 158-159: Please give me more details: The edge of the cup was attached to the pumpkin with glue and a circular piece of sponge? Would you have pictures?

Line 133: The adult parasitoids were kept at 18, 21, 24, 27, 30, 33, and 36°C as well before being eliminated from the cages?

Line 167: The adult parasitoids were kept at a single photoperiod or at different photoperiods before being eliminated from the cages?

Lines 197-219: I think this is more a part of data collection than data analysis. I suggest including this information within the experimental description (lines 107-195).

Lines 200-218: The authors must explain why they used two methods to measure the developmental threshold temperature and effective accumulated temperature.

Line 207: Replace "formular" with "formula".

Line 223: Delete "for".

Lines 229, 232, 235, 240, 241, 245, 247, 251, 253, 255 and all throughout the manuscript: If you are not giving the exact p-value, there is no need to repeat "p<0.05" for every tested parameter because you already gave that information at line 223.

Line 233: Delete the "." after  "30°C".

Lines 234-240: Since it was observed an interaction between sex and temperature, I suggest deleting this part so we can focus on that interaction, as described in the lines 240-242.

Line 239: If you decide to keep this part anyways, please replace "male faster than female" with "males was faster than females".

Lines 247-249: Replace "At 24°C was the highest, which was 55.58, the second-highest female ratio was observed 27°C. There was no significant difference between them, but they were significantly greater than the female ratio at 33°C" with "It was highest at 24ºC, even though there was no significant difference amongst temperatures at the 21-30ºC range".

Lines 250-255: Replace "Temperature significantly influenced the life span of adult S. caerulea in the range of 21-33°C (two-way ANOVA, F (4, 30) = 32.040, p < 0.05). The life span of adult females and males were the longest at 24°C, which were 37.5 d and 38.7 d. Sex did not significantly affect the life span of adult S. caerulea (two-way ANOVA, F (1, 30) = 0.000, p > 0.05). The interaction of temperature and sex did not significantly affect the life span of adult S. caerulea, too (two-way ANOVA, F (4, 30) = 1.948, p > 0.05) (Figure 1f)." with "Temperature significantly influenced the life span of adult S. caerulea at 21-33°C (two-way ANOVA, F (4, 30) = 32.040), but that was not true neither for sex (two-way ANOVA, F (1, 30) = 0.000, p > 0.05) nor for the interaction between temperature and sex (two-way ANOVA, F (4, 30) = 1.948, p > 0.05). The longest lifespan was observed at 24ºC, even though there was no significant difference amongst temperatures at the 21-27ºC range (Figure 1f)."

Lines 258, 261, 262: Replace "temperature of generations" with "degree-days"; delete "ºC-d".

Line 273: Replace "in" with "amongst".

Line 293: Replace "S. caerulea, too" with "S. caerulea either".

Lines 296-297: Figure 3a shows a significant difference between Glucose and Trehalose. Please, correct it.

Line 298: Replace "females, too" with "females either".

Lines 298-299: Figure 3b shows significant differences between Glucose/Honey and Trehalose. Please, correct it.

Lines 300-303: Since it was observed an interaction between sex and nutrition, I suggest deleting this part so we can focus on that interaction, as described in the lines 305-306.

Line 306: Delete "p>0.05".

Line 308 and all throughout the figure description: Delete "The" at the beginning of sentences.

Line 309: Specify if that's the number of eggs/female/day.

Lines 309-312: Delete "by [or] of S. caerulea at different temperatures".

Lines 311-312: Replace "e:" and "f:" with "(e)" and "(f)".

Lines 312-313: Delete "Data in the figure are".

Line 313: Replace "in different" with "amongst".

Line 314: Add a period.

Line 319 and all throughout the figure description: Delete "The" at the beginning of sentences.

Line 320: Specify if that's the number of eggs/female/day.

Lines 320-323: Delete "by [or] of S. caerulea at different light time" and "of S. caerulea at different photoperiods".

Lines 322-323: Replace "e:" and "f:" with "(e)" and "(f)".

Lines 323: Delete "Data in the figure are".

Line 324: Replace "in different" with "amongst".

Line 328 and all throughout the figure description: Delete "The" at the beginning of sentences.

Line 320: Specify that those eggs are laid by each female throughout their entire lifespan.

Lines 328-330: Delete "by [or] of S. caerulea at different supplementary nutrition".

Lines 329: Replace "c:" with "(c)".

Lines 330: Delete "Data in the figure are".

Line 331: Replace "in different" with "amongst".

Lines 338-339: This affirmation is not supported by data (see lines 252-255 and Figure 1f).

Line 340: Replace "rangers" with "ranges".

Line 339: Replace "unde" with "under".

Lines 340-341: If you are going to do this kind of comparison at least pick a parasitoid species that is phylogenetically close to the one that you are studying (e.g., same genus or at least same family). Also, use a range of species, not only one. Finally, use studies that also compare males and females at different temperatures. 

343-345: This affirmation is not supported by your data since 33ºC only significantly differed from 24 and 27ºC (see Figue 1e).

Line 349: This information should be at the Results section.

Line 350: Did you measure both parasitoid and host mortality? Because if you did not, you can't make this statement.

Line 356: Replace " This shows that the oviposition of S. caerulea was promoted by light." with "This shows that oviposition of S. caerulea is positively correlated to the length of the photofase".

Line 359: Replace "This could be because the parasitism rate of this wasp is limited by host density" with "Considering that starting at 14h light the parasitism rate was above 80% independent of the light regimen and consequently had little room to grow, we can infer that the parasitism rate reached a plateau at the regimen of 14h light".

Line 361: What do you mean with "normal life activities"?

Lines 362-363: Comparing results for S. caerulea with those for other parasitoid species especially those from other families is not what we expect in the Discussion section.

Line 377: Replace "ate, such as sucrose…" with "feed on carbohydrate sources such as sucrose…".

Line 382: This statement is not supported by your data. Figure 3ab shows no significant difference between melozitose and any of the other tested nutrients or lack thereof.

Mills, N. (2009). Parasitoids. In V. H. Resh; R. T. Cardé (eds.). Encyclopedia of Insects (2nd ed.). Elsevier. pp. 748–750. ISBN 978-0123741448.

Reviewer 3 Report

The authors have graphed and presented their results clearly, drawing some attention to the implications of their findings. I found the study of interest and a good contribution to the knowledge of bioecology of insect biocontrol agents (BCAs). The methods used are somewhat appropriate for the objectives of the work and, in general, well depicted. The resulting figures are sufficient, informative, and of good quality helping to follow the reasoning throughout the manuscript. I recommend it for publication, but after a major revision that should solve the following manuscript’s constraints:

1) The introduction and discussion provide no insight on how this MS relates to the various other ones cited in the text or concerns that have been raised by other researchers. This article should provide details on all these fronts to provide the proper context for the work. Authors do not present any hypotheses or expectations that could be connected to previous studies; adding these details will improve the paper. This article should provide details on all these fronts to provide the proper context for the work.

2) Unjustified/inappropriate analyses and/or flawed reasoning: Lns:220-225: ANOVA on percentage parasitism data and female ratios? If so, consider an analysis better suited to this response variable, logistic GLM is one, binomial distribution is another one; arsine transformations may not be the best choice (see Warton & Hui’s MS on asinine arcsine transformations of proportions data published in Ecology 2011 92, 3-10). The major difficulty with modelling proportion data is that the responses are strictly bounded. There is no way that the percentage dying can be greater than 100% or less than 0%. But if we use simple techniques such as regression, analysis of variance or covariance, then the fitted model could quite easily predict negative values or values greater than 100%, especially if the variance was high and many of the data were close to 0 or close to 100%. The logistic curve is commonly used to describe data on proportions, because, unlike the straight-line model, it asymptotes at 0 and 1 so that negative proportions and responses of more than 100% cannot be predicted. Briefly, proportions are based on number of cases. Would you give the same weight to a proportion of 2 out of 4 cases (not very reliable) and a more reliable proportion of 20 out of 40 cases? The natural solution is to use the odds and odds ratio, and a binomial distribution to test for change in proportion as a change in the odds, as described in the arcsine asinine publication (see Ecology 2011 92, 3-10). That way you give 50 % of 40 its due, compared to 50% of 4. Results and Discussion sections should be revised accordingly.

3) My other concern is that the authors are extrapolating the applicability of their results beyond what the design supports. With respect to temperature effects on development and reproduction S. caerulea, these are only development and reproduction data from sets of highly artificial constant temperatures ranging from18C-33C, so the inference power of the paper is very limited, but authors do not acknowledge this detail at all and need to be more forthcoming. This is a critical limitation of the study, and the authors must concede and discuss this. It is well known that most of laboratory experiments are conducted under constant temperatures whereas in nature daily temperature fluctuations can be very wide. The interaction of cyclic temperatures with nonlinear function of development and reproduction parameters can introduce significant deviations from the parameters developed here. So, I am suggesting to the authors to tone-down the language a little and admit that there are still substantive uncertainties to be considered, including uncertainty as to how generalizable the results are to open field conditions. This is not to diminish the data gathered in this study, they are of value. But it is important for the authors not to overgeneralize, and to warn the reader including regulatory agencies, against doing so as well.

4) Some of the authors statements would be much stronger if they tie their work to the body of literature that has built up on the bioecology and reproductive biology of mass-produced endo- and ectoparasite insect biocontrol agents (BCAs) for field releases in CBC programs (see my comments under #3). They all point to the same direction and could be connected to this study. Some examples are J. Econ. Entomol. 112: 1560-1574 (mass produced ectoparasite BCAs) or J. Econ. Entomol. 112:1062-1072 (mass produced endoparasite BCAs), but there are others too. These studies provide strong evidence of increased longevity in BCAs reared at non-stressful low temperatures when compared to higher temperature regimes. They further suggest that the parasitism or percentage host mortality was significantly higher at intermediate temperatures than at cline margins. Adding these details will improve the paper.

Good luck!

Round 2

Reviewer 2 Report

This manuscript still needs improvement in the English language (style, grammar, and vocabulary). There are still so many mistakes that are a clear lack of attention (e.g., lines 108 and 112 - the scientific names included the names of the describers - and they were even italicized). The authors did not explain anywhere in the manuscript why they used two methods to measure the developmental threshold temperature and effective accumulated temperature. Sometimes they say that they implemented the suggestions that I made (e.g., Figs. 1f and 2f), but when I go and check I do not see a change. I still think that the Discussion needs considerable improvement as it still consists of repetition from the Results section and comparisons of the authors' results with the literature. Hence, while I do appreciate the efforts of the authors to improve the manuscript, I still have way too many reasons to recommend this manuscript to be rejected for publication in Insects.

Reviewer 3 Report

Authors have done a nice job addressing all of my original comments and those of other reviewers. I have no further suggestions to improve the paper. Thank you.

Author Response

Thank you very much for your comments.